# Dietary Inflammatory Index, Obesity, and the Incidence of Colorectal Cancer: Findings from a Hospital-Based Case-Control Study in Malaysia

**DOI:** 10.3390/nu15040982

**Published:** 2023-02-16

**Authors:** Nor Hamizah Shafiee, Nurul Huda Razalli, Mohd Razif Shahril, Khairul Najmi Muhammad Nawawi, Norfilza Mohd Mokhtar, Ainaa Almardhiyah Abd Rashid, Lydiatul Shima Ashari, Hamid Jan Jan Mohamed, Raja Affendi Raja Ali

**Affiliations:** 1Department of Medicine, Faculty of Medicine, Universiti Kebangsaan Malaysia, Cheras, Kuala Lumpur 56000, Malaysia; 2Dietetics Programme, Faculty of Health Sciences, Universiti Kebangsaan Malaysia, Kuala Lumpur 50300, Malaysia; 3GUT Research Group, Department of Medicine, Faculty of Medicine, Universiti Kebangsaan Malaysia, Cheras, Kuala Lumpur 56000, Malaysia; 4Centre for Healthy Ageing and Wellness (HCARE), Faculty of Health Sciences, Universiti Kebangsaan Malaysia, Kuala Lumpur 50300, Malaysia; 5Gastroenterology Unit, Department of Medicine, Universiti Kebangsaan Malaysia, Cheras, Kuala Lumpur 56000, Malaysia; 6Department of Physiology, Faculty of Medicine, Universiti Kebangsaan Malaysia, Cheras, Kuala Lumpur 56000, Malaysia; 7Nutrition and Dietetics Programme, School of Health Sciences, Universiti Sains Malaysia, Kota Bharu 16150, Malaysia; 8School of Medical and Life Sciences, Sunway University, Petaling Jaya 47500, Malaysia

**Keywords:** diet, chronic inflammation, colorectal cancer risk, inflammatory diet

## Abstract

Obesity-mediated inflammation represents a key connection between the intake of foods with high inflammatory potential and colorectal cancer (CRC) risk. We aimed to explore the association between energy-adjusted dietary inflammatory index (E-DII) in relation to CRC risk in both obese and non-obese subjects. This study included 99 histopathologically confirmed CRC cases, 73 colonic polyps cases, and 141 healthy controls from tertiary medical centres in both urban and suburban areas in Peninsular Malaysia. The subjects were categorised into body mass index (BMI) < 25 kg/m^2^ and BMI ≥ 25 kg/m^2^ groups. E-DII scores were computed based on dietary intake assessed using a validated food frequency questionnaire (FFQ). Logistic regression models were used to estimate odds ratios (ORs) and 95% confidence intervals (CIs), adjusted for potential cofounders. The mean dietary energy intake and mean BMI values of the subjects tended to increase as the E-DII scores increased (*p* for trend < 0.001). E-DII was significantly related to CRC risk only in obese subjects (OR = 1.45; 95% CI = 1.30–1.77; *p* < 0.001 for trend). Stratified analyses of risk factors showed significant associations between E-DII and CRC risk by age group (*p* for interaction = 0.030), smoking status (*p* for interaction = 0.043), and anthropometric indices for both males and females (*p* for interaction < 0.001) in the most pro-inflammatory E-DII quartile vs. the lowest E-DII quartile. Overall, pro-inflammatory diets were associated with an increased incidence of CRC in the Malaysian population, particularly in obese subjects.

## 1. Introduction

Colorectal cancer (CRC) is the third most prevalent gastrointestinal malignancy worldwide, and it was ranked as the second leading cause of cancer-related deaths in 2020 [1]. Malaysia recorded an 11.3% surge in new cancer cases from 103,507 in 2007–2011 to 115,238 in the 2012–2016 period, making CRC the second most common cancer in Malaysia after breast cancer [2]. Excess adiposity, a worldwide pathological condition, has been identified as a risk factor in CRC pathogenesis [3]. It is characterised by low-grade chronic inflammation, which is a major mediator of colorectal pathogenesis [4]. Obesity-induced inflammation, caused by excessive dietary intake, plays an important role in CRC risk and progression, notably through the continuous increased secretion of pro-inflammatory cytokines in the circulation and tissues, such as interleukin (IL)-1, IL-6, IL-8, and tumour necrosis factor-α (TNF-α) [5].

It is generally recognised that specific food items are more likely to trigger chronic gut inflammation, which eventually induces cancer cell proliferation. For instance, a pro-inflammatory western-style diet high in red meat, processed meat, fat, and refined grains may trigger the inflammatory process in the colon that leads to colorectal carcinogenesis, whereas anti-inflammatory diets rich in fruits, vegetables, and fibre may reduce gut inflammation and thus protect against CRC risk [6,7,8,9,10,11]. Therefore, assessing the potential impact of diet on inflammation could aid in the development of dietary strategies to reduce inflammation and the risk of CRC. The dietary inflammatory index (DII), a novel composite score based on a range of nutrients and foods, was developed with a focus on specific biological pathways modulating the impact of dietary factors on inflammation. It is a new approach used to quantify the inflammatory potential related to an individual’s regular eating habits, with scores on a continuum from the most highly pro-inflammatory to the most highly anti-inflammatory [12].

In recent years, published studies have shown that a more inflammatory diet, as indicated by a higher DII score, plays a significant role in colorectal carcinogenesis [13,14,15,16,17,18,19,20,21,22,23,24,25,26,27]. However, most of the relevant studies have been conducted in Western populations. So far, to our knowledge, there have been few pertinent studies evaluating the association between the DII and CRC risk in Asian populations [23,24,25,26,27]. Furthermore, it is unknown whether the inflammatory links between obesity and diet may increase the risk of CRC among Malaysians. To the best of our knowledge, no research has been done to investigate the role of pro-inflammatory diets in the multi-ethnic population in Malaysia, whose dietary patterns mainly consist of rice, which is the staple food, accompanied by generous amounts of fresh seafood, chilies, curries, palm oil, coconut milk, and a variety of spices that differ from those studied in other populations. Therefore, in the present study, we aimed to assess the relationship between the inflammatory impact of diet on CRC risk, as measured by the DII score, through a hospital-based case-control study in two different cities in Peninsular Malaysia, specifically among obese and non-obese individuals.

## 2. Materials and Methods

### 2.1. Study Design and Selection of Subjects

The hospital-based case-control study using purposive sampling was conducted in a tertiary medical centre in two different cities in the central (Kuala Lumpur, KL) and northeastern regions (Kota Bharu) of Peninsular Malaysia, which represent the urban and suburban areas, respectively. KL has the highest population density with 7188 people per square kilometre, while Kota Bharu has a population density of 128 people per square kilometre [28]. The minimum required sample size was calculated with a case to control ratio of 1:2, 80% study power (1 − β), and 5% level of significance (α), assuming that the expected proportions in case and control groups were 0.606 and 0.387, respectively [29]. All subjects who were eligible for the study and agreed to participate provided signed informed consent. The study protocol was reviewed and approved by the Universiti Kebangsaan Malaysia Medical and Research Ethics Committee (UKMREC; FF-2020-005) and the Human Research Ethics Committee of Universiti Sains Malaysia (USM/JEPeM/19060354).

The eligibility criteria included: (1) males and females aged 18 to 80 years (at the time of diagnosis or interview); (2) willingness to cooperate in the study; (3) not being on any special diet that could influence their weight status; and (4) having no previous abdominal surgeries associated with bowel obstruction. The CRC cases were patients with histopathologically confirmed malignancy, diagnosed no more than one year before study enrolment, and had no previous cancer diagnosis at other anatomic sites. Cases of colonic polyps included those with histological tubular adenoma with low-grade dysplasia based on colonoscopic evaluation. Healthy controls were selected from subjects that visited the medical centre for the colonoscopy screening and had normal findings, either no polyps or no cancer diagnosis.

Exclusion criteria included: (1) being pregnant or breastfeeding; (2) having polyps ≥1 cm, tubulovillous adenoma, and high-grade dysplasia; (3) having severe chronic medical illnesses such as neurological diseases, including stroke and muscle disorders, and other cancers; and (4) having a severe internal medical condition requiring dietary restriction such as renal disease, lactose intolerance, or gluten intolerance. The controls were frequency matched on age (±10 years) with cases. Of the 321 subjects who agreed to participate in the study and completed the questionnaire, 4 CRC patients, 1 colonic polyps patient, and 3 healthy controls were excluded due to the implausible total energy intake (<600 or >3500 kcal/day for women, and <800 or >4200 kcal/day for men) [30]. These cut-offs were based on the crude method of excluding the subjects reporting energy intake at the extremes of a range of intake from the analysis. When using a dietary reporting method that cannot provide an accurate estimate of energy intake, such as the food frequency questionnaire (FFQ), then a crude method to account for misreporting may be the best option [31]. Less than 5% of the subjects were ineligible to participate in the study. In total, 313 subjects [n = 99 (CRC patients); n = 73 (colonic polyps); n = 141 (healthy controls)] were included in the final statistical analysis. The selection of subjects is illustrated in Figure 1.

### 2.2. Data Collection

The same structured questionnaire was used at each study site to collect the demographic data and information about the known risk factors for CRC from the eligible subjects during their initial hospital visit through a face-to-face interview with a trained interviewer. The questionnaire included questions regarding age, sex, ethnicity, residence, marital status, employment status, educational level, household income, family history of CRC, personal history of polyps, smoking status, anthropometric indices, and dietary intake.

### 2.3. Anthropometric Measurements 

Anthropometric data were collected using standardised protocols by a trained research dietitian prior to the interview session. Weight and height were measured twice, and the average value was used in the analysis. A portable TANITA digital weighing scale model SC-330 (Tanita Corporation, Tokyo, Japan) was used to measure the subjects’ weight and body fat percentage (BF%) to the nearest 0.1 kg and 0.1%, respectively, while a SECA portable stadiometer model 213 (Seca GmbH & Co. KG., Hamburg, Germany) was used to measure the subjects’ height to the nearest 0.1 cm. The obtained height and weight were used to calculate BMI using the formula, BMI = weight (kg)/height (m^2^). The BMI cut-off point was classified according to the criteria defined by the World Health Organization (WHO) [32]. In this study, we grouped the subjects into two different categories, which are the obese group for those with a BMI ≥ 25 and non-obese group for those with a BMI < 25.

Waist circumference (WC) and hip circumferences (HC) were measured to the nearest 0.1 cm using a non-elastic measuring tape Seca 201 (Vogel and Halke GmbH and Co., Hamburg, Germany). WC was measured at the midpoint between the inferior margin of the last rib and the ilium, while the HC was measured at the point of the largest bulge of the gluteus maximus. The waist–hip ratio (WHR) was then calculated as the ratio of WC (cm) to HC (cm). For the assessment of abdominal obesity, the WC cut-off was classified as ≥90 cm for men and ≥80 cm for women, while men with a WHR ≥ 1.0 and women with a WHR ≥ 0.85 were considered obese, according to the recommended classification from the WHO (2008) [33]. The BF% cut-off point for obesity proposed by the WHO was set at 25% for men and 35% for women [34].

### 2.4. Dietary Assessment

The dietary intake of each subject was assessed using a semi-quantitative food frequency questionnaire (FFQ), which was an adapted version of the FFQ used in the National Health and Morbidity Survey (NHMS) 2014 [35]. The NHMS was a nationally representative food consumption survey conducted in both Peninsular Malaysia and East Malaysia. The FFQ applied in this study was modified to include food sources associated with CRC development. Details on the validation and reliability study of this FFQ are provided elsewhere [36]. The finalised FFQ used in this study includes 142 food items from 13 food groups: cereal products, meats, fish and seafood, eggs, vegetables, nuts and legumes, milk and dairy products, condiments, bread spread, fruits, confectionaries, fast food, and sugar-sweetened drinks. 

The FFQ was administered through a face-to-face interview in which subjects were requested to provide the type of normally consumed food items, the frequency of intake of each food item based on the standard serving size, and the number of serving sizes consumed in the preceding year prior to diagnosis/interview using five response categories (“never,” “per day,” “per week,” “per month,” or “per year”), guided by a trained dietitian. Each food item in the FFQ was assigned a portion size using common household units, such as spoons, bowls, cups, bowls, matchbox sizes, glasses, and plates, to estimate the serving size of the food eaten. To minimise errors while assessing the dietary intake of subjects, the interview process was started by asking the subject to recall all the food items that were typically consumed daily, and the recall process was built up over weeks and months. Further, the completed FFQs were cross-checked by a dietitian for completeness and accuracy in terms of portion size and ingredients recorded. 

Based on the FFQ data, reported food intakes in grams (g) per day were calculated for each subject. Nutrient and total energy intakes per day were estimated using the Nutrient Composition of Malaysian Foods, Malaysian Atlas of Food Exchanges and Portion Sizes, Album Makanan Malaysia, and Malaysian Food Composition Database (MyFCD, 2020) [37,38,39,40]. For food items that were not available in the published literature mentioned above, nutritional food labels and recipes from websites were used as references. The amount of daily food intake was calculated from the FFQ according to the following formula: {frequency of intake (the conversion factor) × serving size × total number of servings × weight of food in one serving (g)} [41]. From the values of the amount of food consumed per day, the detailed analysis of the intake of nutrients was calculated using the Nutritionist Pro™ Diet Analysis Software version 7.8.0 (Axxya Systems, version 2020, Redmond, WA, USA) to obtain energy and nutrient values for each subject.

### 2.5. Dietary Inflammatory Index (DII) Score

The FFQ-derived dietary data was used to calculate the energy-adjusted DII (E-DII) scores for each of the enrolled subjects. The comprehensive description of development [12] and construct validation [42] of the DII are available elsewhere. Briefly, 1943 research articles published through 2010 that reported the link between 45 food parameters (consisting of whole foods, nutrients, and flavonoids) and six inflammatory biomarkers [IL-1β, IL-4, IL-6, IL-10, TNF-α, and C-reactive protein (CRP)] were reviewed and scored. Regionally representative datasets based on diet surveys from 11 countries were collectively used as comparative standards for each of the 45 parameters. To calculate the E-DII scores for the subjects of this study, the intake scores of the noted datasets were applied. Food and nutrient intake from the FFQ were first adjusted for total energy intake (density method = nutrient/total energy intake × 1000 kcal) and then standardised by creating a z-score for each food component using mean and standard deviation (SD) values from a global energy-adjusted database. The energy-adjusted standardised dietary intake was then multiplied by the respective literature-derived inflammatory effect score to obtain a food parameter-specific E-DII score for an individual, and summed across all components to obtain the overall E-DII score for each study subject.

The resulting E-DII score increases with the increased inflammatory potential of the diet, with a higher or more positive value of the E-DII score indicating a more pro-inflammatory diet, whereas a lower or more negative value of the E-DII score represents a more anti-inflammatory diet. For the current study, 29 food parameters were included to calculate the E-DII score: energy, carbohydrates, proteins, total fat, saturated fatty acids (SFA), trans fat, monounsaturated fatty acids (MUFA), polyunsaturated fatty acids (PUFA), alcohol, vitamin A, C, D, E, B6, B12, β-carotene, caffeine, cholesterol, dietary fibre, folic acid, iron, magnesium, niacin, omega-3 fatty acids, omega-6 fatty acids, riboflavin, selenium, thiamin, and zinc. For analytical purposes, E-DII scores were categorised into four groups (quartiles) to investigate the relationship with the different variables. The lowest and highest E-DII scores were found in the first and fourth quartiles, respectively.

### 2.6. Statistical Analysis

The collected data was analysed using the IBM SPSS for Windows, version 20.0 (IBM, Armonk, NY, USA). The distribution of all variables was verified for normality using the Kolmogorov–Smirnoff test, with verification using skewness and kurtosis. Continuous and categorical variables were expressed as means with standard deviation (SD) and frequency as percentages (%), respectively. The Jonckheere–Terpstra test and the Mantel–Haenszel Chi-square test were used to evaluate trends in *p* values for continuous and categorical data, respectively. Analysis of variance (ANOVA) was used to test for a mean difference in the distribution of the dietary intake and specific food groups across the E-DII quartiles. A multiple covariate-adjusted logistic regression model was applied to generate the OR and corresponding 95% confidence interval (95% CI) for the risk of CRC and colonic polyps according to E-DII quartile. The multivariable model was adjusted for BMI (for all subjects), age, sex, and smoking status. A linear test for trend was conducted by including the median value of each E-DII quartile as a continuous variable in the regression model. Stratified analyses were carried out in subgroups of factors that have been linked with CRC risk, including sex, age groups, smoking status, and anthropometric indices. The lowest E-DII quartile (the most anti-inflammatory diet) was the reference group for all models. All statistical tests were run with an alpha level of 0.05. 

## 3. Results

The demographic and other characteristics of the 313 eligible subjects based on the E-DII quartile distribution are shown in Table 1. In this study, the E-DII scores ranged from −2.45 (maximum anti-inflammatory) to +4.51 (maximum pro-inflammatory). The controls reported a more anti-inflammatory diet than the corresponding cases of CRC and colonic polyps (Figure 2). The proportion of CRC subjects increased across the E-DII quartile, while healthy subjects were more prevalent in the lower E-DII quartile (*p* = 0.004). The mean age of the subjects increased as the E-DII increased (*p* < 0.001), with subjects in the higher quartile being older (age ≥ 50 years old) than those in the lower quartile (*p* < 0.001). In regards to ethnicity, subjects in the highest E-DII quartile (Q4) were mostly Malays, while Chinese and Indians were less prevalent (*p* < 0.001). The proportion of subjects who were employed decreased as E-DII increased (*p* < 0.001). The educational level also decreased as E-DII increased (*p* < 0.001). There were more subjects with a bachelor’s degree or higher, while fewer subjects had just completed their primary education. The proportion of subjects living in urban areas increased across the E-DII quartile, whereas subjects living in suburban areas were more prevalent in the lower E-DII quartile (*p* < 0.001). The family history of CRC and personal history of polyps did not show any notable tendency across the E-DII quartile. Meanwhile, as the E-DII score grew, subjects’ mean dietary energy intake and mean BMI values tended to rise as well (*p* < 0.001). Subjects in Q4 were more likely to be obese than those in the lower quartile, whereas non-obese subjects were more likely to be in the lowest E-DII quartile (Q1) (*p* < 0.001). Furthermore, the mean WC, WHR, and BF% values increased as the E-DII score increased for both males and females (*p* < 0.05). 

The distribution of 29 food parameters across the E-DII quartile based on BMI categories is presented in Appendix A. In both obese and non-obese subgroups, higher E-DII scores were significantly related to lower intakes of dietary fibre, ω-3 fatty acids, PUFA, thiamin, riboflavin, niacin, vitamin B6, vitamin C, vitamin E, beta-carotene, magnesium, and zinc (*p* < 0.05). Subjects in Q4 had the highest intake of carbohydrates, protein, total fat, iron, and SFA, in both obese and non-obese subjects (*p* < 0.001). The reported intake of vitamin A and selenium was significantly lower in the higher E-DII quartile compared to those in the lower quartile, while trans fat showed a significant increase across the E-DII quartiles only for obese subjects (*p* < 0.001). Meanwhile, the reported intake of folic acid was significantly higher in Q4 than in the other quartiles only in non-obese subjects (*p* = 0.004). The distribution of certain food groups across the quartile of the E-DII was also examined in both obese and non-obese subjects (Appendix A). Concerning the distribution of various food groups, cereal products, sugar-sweetened beverages, condiments, and fast foods showed a significant increase across the E-DII quartiles for both BMI categories (*p* < 0.05), while a higher E-DII score was associated with a higher intake of meat and confectionary only in obese subjects (*p* < 0.001). The intake of fruits and vegetables decreased significantly across the E-DII quartiles in both obese and non-obese subjects (*p* < 0.001). Meanwhile, intake of nuts and legumes decreased significantly across the E-DII quartiles only in obese subjects (*p* = 0.018). 

In the analysis of CRC incidence (Table 2), a more pro-inflammatory diet (Q4 vs. Q1) was associated with an increased risk of CRC (OR = 1.46; 95% CI = 1.32–1.80; *p* < 0.001 for trend) for all subjects. When the subgroups were defined by BMI categories, there were significant differences in the association between the dietary inflammatory potential and CRC risk for obese subjects (*p* < 0.001 for trend). When comparing subjects in Q4 vs. Q1, higher E-DII scores were associated with a 45% (OR = 1.45; 95% CI = 1.30–1.77) higher risk of developing CRC in obese subjects, adjusted for age, sex, and smoking status. Table 3 provides the OR and 95% CIs for the risk of colonic polyps according to E-DII quartile. For all subjects, there was no significant association with the risk of colonic polyps (OR = 0.65; 95% CI = 0.56–1.15; *p* = 0.230 for trend). When comparing subjects in Q4 vs. Q1, subgroup analysis based on BMI categories also revealed no significant association with the risk of colonic polyps.

Stratified analyses by subgroups of risk factors (Table 4) showed statistically significant interaction between E-DII and CRC risk by age group (*p* for interaction = 0.030), smoking status (*p* for interaction = 0.043), and anthropometric indices (WC, WHR, and BF%) for both men and women (*p* for interaction < 0.001). When comparing the highest vs. lowest quartile, stronger positive associations were observed among subjects who were 50 years and older (OR = 2.09; 95% CI = 1.40–1.85), ever smokers (OR = 1.98; 95% CI = 1.55–2.03), WC ≥ 90 cm for males (OR = 2.28; 95% CI = 1.96–2.43), WC ≥ 80 cm for females (OR = 2.48; 95% CI = 2.16–2.67), WHR ≥ 1.0 for males (OR = 2.33 95% CI = 2.06–2.53), WHR ≥ 0.85 for females (OR = 2.58; 95% CI = 2.26–2.73), BF ≥ 25% for males (OR = 2.52; 95% CI = 2.27–2.74), and BF ≥ 35% for females (OR = 2.47; 95% CI = 2.13–2.61). 

## 4. Discussion

The current findings provide the first evidence of a link between higher E-DII scores, which reflect dietary intakes with greater inflammatory potential, and an elevated risk of CRC in a multi-ethnic Malaysian population. The findings of this population, whose dietary choices differ markedly from those of the more well-studied in the West, reaffirm the use of the DII as a tool for relating the inflammatory potential of diet to CRC in a wide range of populations. Our findings add to the evidence that higher DII scores are associated with an increased incidence of CRC in the United States [13,14,15,16,17,18,19], Europe [20,21,22], and Asia [23,24,25,26,27]. A meta-analysis with nine studies reported a 40% increase in CRC risk (relative risk (RR) = 1.40; 95% CI = 1.26–1.55; *p* < 0.001), when comparing the highest vs. lowest (reference) DII categories [43], confirming that a pro-inflammatory diet is linked with an increased risk of CRC. In recent years, a more pro-inflammatory diet has also been implicated in the occurrence and development of different types of other cancers, including urologic cancers [44,45,46,47], upper aerodigestive tract cancers [48,49,50,51,52,53], breast cancer [54,55,56,57,58], gastric cancer [59,60], endometrium cancer [61], and lung cancer [62]. In the overall analysis, significant associations were found both in case-control [44,45,48,49,50,51,52,53,54,56,57,59,60,61] and cohort studies [46,47,55,58,62] to support a role for dietary inflammation in the pathogenesis of cancer. These findings highlight the potential benefits of adopting a healthier dietary pattern to reduce the risk of cancer.

In this study, we found that those with higher E-DII scores had more unhealthy diets (pro-inflammatory diets), while those with lower E-DII scores consumed more healthy diets (anti-inflammatory diets). In general, our findings indicate that the dietary components that associated with CRC risk are consistent with their inflammatory potential [12]. According to a recent study, adherence to the Western dietary pattern was substantially related to overall CRC risk (OR = 1.50; 95% CI = 1.20–1.87), whereas adherence to the Mediterranean diet was inversely associated (OR = 0.65; 95% CI = 0.53–0.80) [63]. There are numerous pathways that potentially explain the role of diet-mediated inflammation in CRC pathogenesis. First, the intake of pro-inflammatory diets can increase insulin resistance or glucose intolerance by increasing systemic inflammation [64], resulting in continuously elevated circulating levels of insulin, glucose, and triglycerides, which may exert growth-promoting effects on colonic epithelial cells and potentially expose them to reactive oxygen species (ROS), thereby increasing the risk of CRC [65,66]. Pro-inflammatory diets also reduce the release of IL-10 and increase intestinal permeability due to intestinal epithelial barrier dysfunction, which allows microbial translocation into the mucosa and may induce low-grade inflammation [67,68]. Another possible mechanism is through the progressive increase in the expression of the cyclooxygenase-2 (COX-2) enzyme in colonic epithelial cells, resulting in local inflammation and oxidative stress [69], which may subsequently induce focal proliferation and mutagenesis in the colon [14,66]. COX-2 is stimulated by inflammatory cytokines and growth factors, and inhibited by anti-inflammatory components of DII [70].

In the subgroup analysis, we found significant associations only in obese subjects compared to lean subjects, suggesting that the association may be reinforcing obesity-mediated inflammation. A recent systematic review and meta-analysis of twelve studies reported a higher risk of developing CRC in overweight and obese subjects (BMI ≥ 25 kg/m^2^) compared with normal weight subjects (OR = 1.42; 95% CI = 1.19–1.68) [71]. Furthermore, after adjusting for cofounders, we also observed a stronger association between central obesity (as measured by WC and WHR) as well as BF%, and the risk of CRC. One proposed mechanism underlying the link between obesity, inflammation, and CRC risk is through the secretion of inflammatory cytokines such as TNF-α, IL-1β, IL-6, and monocyte chemoattractant protein (MCP)-1 by adipose tissue-derived cells [72]. Obese adipose tissue is infiltrated by M1-polarised macrophages (pro-inflammatory phenotype), resulting in the high secretion of nitric oxide (NO), which can induce DNA damage and thereby facilitate tumourigenesis [72]. Meanwhile, lean adipose tissue has an activated M2 macrophage phenotype, which is implicated in tissue repair and the prevention of inflammation [72]. 

Adipose-derived hormones (adipokines) have also been implicated as potential mediators of the link between obesity and CRC risk. The dysregulation of adiponectin and leptin, two of the most abundant adipokines during obesity, is linked to a pro-inflammatory response in CRC [4]. Leptin levels are elevated in obese subjects, which stimulates cell growth, proliferation, migration, and invasion by activating the leptin receptor and transcription 3 (STAT3) pathway, which modulates extracellular signal-regulated kinase (ERK) signalling to increase inflammatory cytokines, promote tumour vascularization, and inhibit apoptosis [4]. Adiponectin, unlike leptin, decreases with increased adiposity and may suppress TNF-α production by macrophages, while also inhibiting TNF-α -induced expression of adhesion molecules through nuclear factor kappa light chain-enhancer of activated B cells (NF-κB) signalling pathways [4]. 

Other risk factors may affect the role played by diet-associated inflammation in colorectal carcinogenesis. We found stronger association among subjects who were older and smokers. The ageing trend in Malaysia may increase the prevalence of risk factors for CRC, as 80% of CRC cases in Malaysia were diagnosed in older people after the age of 50 [73]. In this study, the E-DII scores increased with age, suggesting that some changes in dietary habits observed in the older subjects may reduce the anti-inflammatory effect of diet compared with younger subjects. Cigarette smoking is linked with a wide range of alterations in inflammatory marker levels, and smoking-induced inflammation may be a critical mechanism in CRC development. Cigarette smoke contains numerous carcinogens, including polycyclic aromatic hydrocarbons, nitrosamines, heterocyclic amines, aromatic amines, and benzene [74] which induce colon tumourigenesis through the activation of nicotinic acetylcholine receptors, formation of DNA adducts, stimulation of tumour angiogenesis, as well as genetic mutation [75].

In view of the fact that the findings of our study indicate the importance of diet-induced inflammation in the development of CRC, several dietary recommendations suggest that encouraging the intake of more anti-inflammatory diets, such as plant-based foods that include a large variety of fruits, vegetables, nuts, and legumes, as well as reducing the intake of pro-inflammatory diets, such as sweetened beverages, confectionaries, and fast food, which contain excessive calories, sodium, saturated fat, and trans-fatty acids, may be a strategy for reducing the risk of CRC. In addition, recommending a diet high in omega-3 fatty acids, vitamin A, vitamin B group, vitamin C, vitamin E, beta-carotene, and minerals like zinc, magnesium, and selenium, which have lower pro-inflammatory scores and antioxidant properties, may help reduce the risk of CRC. Antioxidants can inhibit cancer initiation and progression through a reduction in the formation of free radicals and ROS [76].

The limitations of our study should also be considered. First, this study was prone to information and selection bias. To minimise the information bias, the questionnaires were filled out and further checked through the direct interview by the same trained dietitians, under similar conditions in a hospital setting, to ensure their reliability. To increase the feasibility of the study, subjects were recruited from both urban and rural areas. To reduce selection bias, we recruited subjects who had recently been diagnosed with CRC or colonic polyps. Meanwhile, healthy controls also underwent the same diagnostic/screening procedures to rule out a CRC positive status, rather than their CRC status being self-reported. Furthermore, the high participation rate (approximately 95%) of eligible subjects who were approached to participate reduced the selection bias. Second, memory bias could also be one of the limitations of this study. To minimise this effect, detailed information was collected during the one year prior to diagnosis or the date of interview. The use of valid and reliable FFQs can also help in minimising memory bias. Third, our FFQ did not provide data for the other 16 food parameters for a complete E-DII calculation, such as saffron, thyme/oregano, and rosemary. However, because these components are typically consumed in small amounts or not at all, their absence may not have a significant impact on the scoring. Furthermore, a previous study had computed the DII score with as few as 18 components [27], so the 29 items provided by our FFQ were well above the minimum needed to calculate E-DII scores.

## 5. Conclusions

This study fills the gap by providing new evidence on culture-specific dietary patterns in Malaysia that supports the association between intake of diets with greater pro-inflammatory potential and an increased incidence of CRC, particularly in obese subjects. Our results emphasise the importance of consuming more anti-inflammatory and fewer pro-inflammatory diets to reduce the CRC risk. Improving our understanding of the association between obesity risk factors and CRC carcinogenic processes can be useful for more specific therapeutic approaches for obesity-related CRC treatment in the future. We may offer some recommendations for clinicians to consider when counselling patients who may be interested in modifying their diet for CRC prevention. Future large prospective studies are warranted to confirm these findings.

## Figures and Tables

**Figure 1 nutrients-15-00982-f001:**
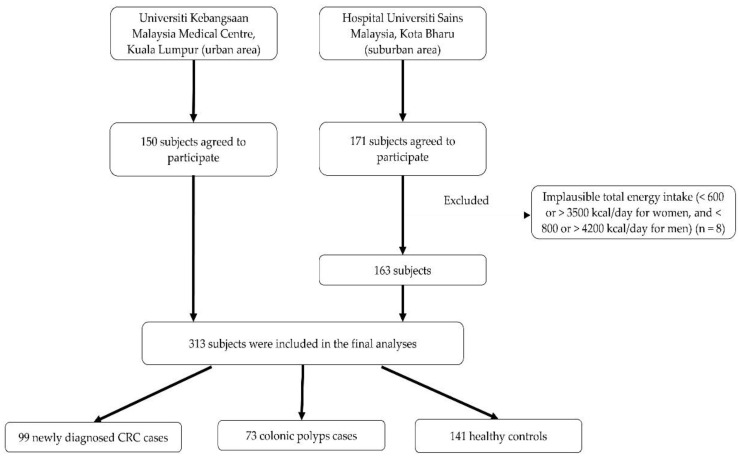
Flowchart for the sample selection. CRC; colorectal cancer.

**Figure 2 nutrients-15-00982-f002:**
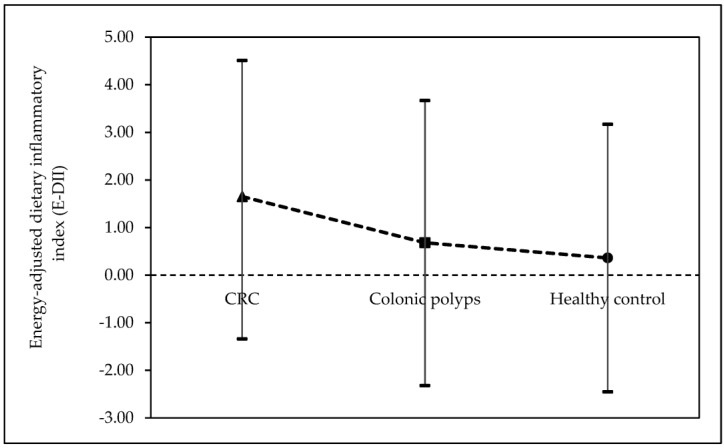
Mean dietary inflammatory index levels in CRC cases (▲), colonic polyps cases (■), and healthy controls (●). CRC; colorectal cancer.

**Table 1 nutrients-15-00982-t001:** Characteristics of the study subjects according to quartile of energy-adjusted dietary inflammatory index (E-DII) score.

Characteristics	Quartiles of E-DII				*p* for Trend ^a^
	Q1 (≤−0.04) Mean ± SD (−0.75 ± 0.65) n = 78	Q2 (−0.04–1.15) Mean ± SD (0.51 ± 0.39) n = 79	Q3 (1.15–2.54) Mean ± SD (1.82 ± 0.43) n = 79	Q4 (≥2.54) Mean ± SD (3.20 ± 0.48) n = 77	
Subjects					0.004 **
CRC	18 (23.1)	25 (31.6)	24 (30.4)	32 (41.6)	
Colonic Polyps	11 (14.1)	24 (30.4)	22 (27.8)	16 (20.8)	
Healthy Control	49 (62.8)	30 (38.0)	33 (41.8)	29 (37.7)	
Sex					0.071
Female	42 (53.8)	41 (51.9)	34 (43.0)	32 (41.6)	
Male	36 (46.2)	38 (48.1)	45 (57.0)	45 (58.4)	
Age (yr) ^b^	49.76 ± 15.2	51.59 ± 14.5	54.27 ± 14.8	58.74 ± 15.3	<0.001
Age groups (yr)					<0.001
<50	40 (51.3)	36 (45.6)	26 (32.9)	18 (23.4)	
≥50	38 (48.7)	43 (54.4)	53 (67.1)	59 (76.6)	
Dietary energy intake (kcal) ^b^	1534 ± 453	1958 ± 309	2085 ± 301	2185 ± 346	<0.001
Ethnicity					<0.001
Malay	40 (51.3)	50 (63.3)	50 (63.3)	64 (83.1)	
Chinese	20 (25.6)	15 (19.0)	22 (27.8)	12 (15.6)	
Indian	18 (23.1)	14 (17.7)	7 (8.9)	1 (1.3)	
Educational Level					<0.001
Middle school or less	9 (11.5)	5 (6.3)	8 (10.1)	15 (19.5)	
High school	22 (28.2)	34 (43.0)	39 (49.4)	41 (53.2)	
College/University	47 (60.3)	40 (50.6)	32 (40.5)	21 (27.3)	
Residence					<0.001
Urban	18 (23.1)	30 (38.0)	43 (54.4)	72 (93.5)	
Suburban	60 (76.9)	49 (62.0)	36 (45.6)	5 (6.5)	
Family history of CRC	26 (33.3)	26 (32.9)	28 (35.4)	20 (26.3)	0.443
Previous history of polyps	9 (11.5)	9 (11.4)	7 (8.9)	4 (5.2)	0.144
BMI (kg/m^2^) ^b^	22.20 ± 3.0	24.66 ± 4.7	25.24 ± 4.7	28.53 ± 4.9	<0.001
BMI categories					<0.001
Non-obese (<25 kg/m^2^)	50 (64.1)	44 (55.7)	34 (43.0)	29 (37.7)	
Obese (≥25 kg/m^2^)	28 (35.9)	35 (44.3)	45 (57.0)	48 (62.3)	
Waist circumference (cm), males ^b^	88.99 ± 6.1	92.95 ± 11.3	96.06 ± 11.4	105.76 ± 11.9	0.002 **
Waist circumference (cm), females ^b^	77.55 ± 7.8	89.54 ± 16.8	89.33 ± 15.4	93.91 ± 17.3	0.005 **
Waist–hip ratio, males ^b^	0.91 ± 0.1	0.92 ± 0.1	0.94 ± 0.1	1.01 ± 0.1	<0.001
Waist–hip ratio, females ^b^	0.78 ± 0.1	0.83 ± 0.1	0.87 ± 0.1	0.92 ± 0.1	<0.001
Body fat (%), males ^b^	22.99 ± 8.3	25.16 ± 6.8	27.60 ± 5.8	31.76 ± 7.1	<0.001
Body fat (%), females ^b^	28.25 ± 5.9	34.12 ± 8.2	34.60 ± 8.2	35.21 ± 9.2	0.030 **
Employment status					<0.001
Employed	60 (76.9)	55 (69.6)	42 (53.2)	39 (50.6)	
Unemployed	18 (23.1)	24 (30.4)	37 (46.8)	38 (49.4)	
Monthly household income (MYR) ^c^					0.485
<3860	43 (55.8)	37 (47.4)	51 (64.6)	48 (63.2)	
3860–8319	31 (40.3)	37 (47.4)	20 (25.3)	22 (28.9)	
≥8320	3 (3.9)	5 (5.2)	8 (10.1)	7 (7.9)	
Marital Status					0.291
Married	48 (61.5)	49 (62.0)	52 (65.8)	53 (68.8)	
Single/Divorced/Widowed	30 (38.5)	30 (38.0)	27 (34.2)	24 (31.2)	
Smoking Status					0.449
Non-smoker	45 (57.7)	45 (57.0)	47 (59.5)	44 (57.1)	
Former smoker	15 (19.2)	19 (24.1)	19 (24.1)	22 (28.6)	
Current smoker	18 (23.1)	15 (19.0)	13 (16.5)	11 (14.3)	

E-DII, energy-adjusted dietary inflammatory index; CRC, colorectal cancer; BMI, body mass index; Q, quartile. Q1 refers to scores indicating the most anti-inflammatory diet (lowest E-DII), and Q4 refers to scores indicating the most pro-inflammatory diet (highest E-DII). The data of continuous variables are presented as mean ± standard deviation, and the data of categorical variables are presented as frequency number (%). ^a^ Mantel–Haenszel Chi-square test for categorical data. ^b^ Jonckheere–Terpstra test for continuous data. ^c^ Based on the cut-off of Eleventh Malaysia Plan (2015). 1 USD = MYR4.40. ** *p* for trend < 0.05 were considered as statistically significant.

**Table 2 nutrients-15-00982-t002:** Odds ratios (ORs) and 95% confidence intervals (CIs) for CRC risk by quartile of the E-DII score for all the subjects and by BMI categories.

Quartile of E-DII Score		All Subjects			BMI < 25 kg/m^2^			BMI ≥ 25 kg/m^2^	
	No. Controls/CRC Cases	Crude OR (95% CI)	Adjusted OR (95% CI) ^d^	No. Controls/ CRC Cases	Crude OR (95% CI)	Adjusted OR (95% CI) ^d^	No. Controls/ CRC Cases	Crude OR (95% CI)	Adjusted OR (95% CI) ^d^
Q1	49/18	1.0 (ref.)	1.0 (ref.)	28/12	1.0 (ref.)	1.0 (ref.)	21/6	1.0 (ref.)	1.0 (ref.)
Q2	30/25	1.03 (0.95–1.40)	1.05 (0.97–1.43)	12/15	1.08 (0.98–1.45)	1.16 (0.92–1.51)	18/10	1.10 (1.02–1.49)	1.06 (1.02–1.41)
Q3	33/24	1.26 (1.16–1.63)	1.30 (1.23–1.72)	9/17	1.39 (0.96–1.66)	1.37 (0.96–1.55)	24/7	1.20 (1.10–1.55)	1.10 (1.08–1.46)
Q4	29/32	1.39 (1.27–1.74)	1.46 (1.32–1.80)	13/10	1.34 (0.99–1.63)	1.32 (0.87–1.46)	16/22	1.36 (1.25–1.77)	1.45 (1.30–1.77)
*p* for trend		<0.001	<0.001		0.130	0.182		<0.001	<0.001

E-DII, energy-adjusted dietary inflammatory index; BMI, body mass index; Q, quartile; OR, odds ratio; CI, confidence interval; ref., reference. Q1 refers to scores indicating the most anti-inflammatory diet, and Q4 refers to scores indicating the most pro-inflammatory diet. ^d^ adjusted for BMI (for all subjects), age, sex, and smoking status. *p* < 0.05 was considered as statistically significant. *p* for trend when E-DII used as a continuous variable.

**Table 3 nutrients-15-00982-t003:** Odds ratios (ORs) and 95% confidence intervals (CIs) for colonic polyps risk by quartile of the E-DII score for all the subjects and by BMI categories.

Quartile of E-DII Score		All subjects			BMI < 25 kg/m^2^			BMI ≥ 25 kg/m^2^	
	No. Controls/ Colonic Polyps Cases	Crude OR (95% CI)	Adjusted OR (95% CI) ^d^	No. Controls/ Colonic Polyps Cases	Crude OR (95% CI)	Adjusted OR (95% CI) ^d^	No. Controls/ Colonic Polyps Cases	Crude OR (95% CI)	Adjusted OR (95% CI) ^d^
Q1	48/11	1.0 (ref.)	1.0 (ref.)	28/10	1.0 (ref.)	1.0 (ref.)	20/1	1.0 (ref.)	1.0 (ref.)
Q2	30/25	0.93 (0.83–1.43)	0.86 (0.76–1.35)	12/17	0.99 (0.89–1.45)	1.82 (0.73–1.32)	18/8	1.12 (0.89–1.50)	1.09 (0.79–1.39)
Q3	33/22	1.05 (0.96–1.53)	0.79 (0.65–1.23)	9/8	1.09 (0.99–1.56)	1.07 (0.97–1.57)	24/14	1.23 (0.93–1.53)	1.24 (0.74–1.36)
Q4	29/16	1.11 (0.99–1.64)	0.65 (0.56–1.15)	13/6	1.14 (0.87–1.62)	1.12 (0.99–1.59)	16/10	1.19 (0.95–1.57)	1.10 (0.86–1.47)
*p* for trend		0.150	0.230		0.130	0.180		0.145	0.160

E-DII, energy-adjusted dietary inflammatory index; BMI, body mass index; Q, quartile; OR, odds ratio; CI, confidence interval; ref., reference. Q1 refers to scores indicating the most anti-inflammatory diet, and Q4 refers to scores indicating the most pro-inflammatory diet. ^d^ adjusted for BMI (for all subjects), age, sex, and smoking status. *p* < 0.05 was considered as statistically significant. *p* for trend when E-DII used as a continuous variable.

**Table 4 nutrients-15-00982-t004:** Odds ratios and 95% confidence intervals for CRC risk by quartile of the E-DII score, stratified by selected factors.

	Quartile of E-DII Score								*p* for Interaction
	Q1		Q2		Q3		Q4		
	No. Controls/CRC Cases	OR (95% CI) ^e^	No. Controls/CRC Cases	OR (95% CI) ^e^	No. Controls/CRC Cases	OR (95% CI) ^e^	No. Controls/CRC Cases	OR (95% CI) ^e^	
Sex									0.272
Female	30/7	1.0 (ref.)	20/12	1.05 (0.92–1.37)	12/13	1.08 (0.95–1.40)	10/13	1.12 (0.99–1.44)	
Male	19/11	1.0 (ref.)	10/13	1.19 (0.96–1.41)	21/11	1.15 (0.93–1.38)	19/19	1.18 (0.95–1.40)	
Age group									0.030 **
<50 years old	24/8	1.0 (ref.)	17/9	1.04 (1.02–1.50)	15/3	1.12 (1.00–1.45)	10/1	1.27 (1.15–1.65)	
≥50 years old	25/10	1.0 (ref.)	13/16	1.29 (1.01–1.43)	18/21	1.49 (1.35–1.80)	19/31	2.09 (1.40–1.85)	
Smoking status									0.043 **
Non-smokers	30/11	1.0 (ref.)	19/17	1.06 (0.93–1.40)	20/13	1.12 (1.00–1.47)	15/18	1.25 (1.13–1.60)	
Ever smokers	19/7	1.0 (ref.)	11/8	1.23 (1.13–1.58)	13/11	1.45 (1.30–1.78)	14/14	1.98 (1.55–2.03)	
Waist circumference (cm), males									<0.001
WC < 90	4/9	1.0 (ref.)	2/7	1.05 (0.92–1.37)	11/2	1.02 (0.89–1.35)	15/6	1.15 (1.02–1.47)	
WC ≥ 90	15/2	1.0 (ref.)	8/6	1.26 (1.17–1.64)	10/9	1.95 (1.50–1.97)	4/13	2.28 (1.96–2.43)	
Waist circumference (cm), females									<0.001
WC < 80	12/4	1.0 (ref.)	8/4	1.09 (0.96–1.43)	10/3	1.22 (1.10–1.57)	7/2	1.48 (1.31–1.78)	
WC ≥ 80	18/3	1.0 (ref.)	12/8	1.31 (1.21–1.67)	2/10	1.67 (1.37–1.87)	3/11	2.48 (2.16–2.67)	
Waist–hip ratio, males									
WHR < 1.0	6/3	1.0 (ref.)	10/8	1.10 (0.98–1.47)	18/10	1.12 (1.01–1.49)	16/16	1.25 (1.16–1.63)	
WHR ≥ 1.0	13/8	1.0 (ref.)	0/5	1.39 (1.28–1.75)	3/1	1.95 (1.51–1.97)	3/3	2.33 (2.06–2.53)	
Waist–hip ratio, females									<0.001
WHR < 0.85	11/4	1.0 (ref.)	8/4	1.15 (1.05–1.57)	9/3	1.19 (1.10–1.57)	8/2	1.21 (1.19–1.67)	
WHR ≥ 0.85	19/3	1.0 (ref.)	12/8	1.26 (1.17–1.67)	3/10	1.65 (1.33–1.80)	2/11	2.58 (2.26–2.73)	
Body fat (%), males									<0.001
BF < 25%	4/3	1.0 (ref.)	10/8	1.18 (1.08–1.57)	12/9	1.21 (1.20–1.67)	10/13	1.35 (1.26–1.73)	
BF ≥ 25%	15/8	1.0 (ref.)	0/5	1.35 (1.27–1.75)	9/2	1.75 (1.42–1.87)	9/6	2.52 (2.27–2.74)	
Body fat (%), females									<0.001
BF < 35%	9/3	1.0 (ref.)	6/3	1.20 (1.12–1.60)	6/2	1.24 (1.22–1.69)	8/0	1.39 (1.28–1.76)	
BF ≥ 35%	21/4	1.0 (ref.)	14/9	1.40 (1.34–1.87)	6/11	1.99 (1.54–2.01)	2/13	2.47 (2.13–2.61)	

E-DII, energy-adjusted dietary inflammatory index; WC, waist circumference; WHR, waist–hip-ratio; BF%, body fat percentage; CI, confidence interval; OR, odds ratio; ref., reference; Q, quartile. ^e^ adjusted for age, sex, BMI, educational level, household income, and family history of CRC. ** *p* < 0.05 was considered as statistically significant.

## Data Availability

The datasets used and/or analyzed during the current study are available from the corresponding author on reasonable request.

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
