# Peer review of "Dietary Inflammatory Index, Obesity, and the Incidence of Colorectal Cancer: Findings from a Hospital-Based Case-Control Study in Malaysia"

_nutrients, 2023, doi:10.3390/nu15040982_

Round 1

Reviewer 1 Report

Shafiee and colleagues investigated the association of colorectal cancer (CRC) risk to energy-adjusted dietary inflammatory index (E-DII) in Malaysian population. The study recruited 99 confirmed CRC individuals, 73 colonic polyps cases and 141 healthy people as a control. All subjects underwent substantial assessment and analysis. The final data allowed authors to firmly conclude that pro-inflammatory diets are associated with an increased incidence of CRC in the Malaysian population. Even similar association has been established before in Western population, this study added important evidence to reaffirm this association. Besides, authors found that the association between pro-inflammatory diet and CRC risk is more pronounced in obese subjects. They also identified smoking as a strong factor related to increased CRC risk. Overall, this is a well-written and important study. The conclusion is well supported by the analysis and data presentation. I have a few points below for the authors to consider before the manuscript can be accepted for publication.

1.       In the Introduction line 71, the authors state that the evaluation of DII and CRC risk is lack in Asian populations, but in the Discussion line 351, the association is claimed to be revealed in Asian populations (with four references cited). This inconsistent statement needs to be reconciled.

2.       Is there any study exploring relationship between pro-inflammatory diets and other cancer type? If yes, it should be discussed in the manuscript.

3.       It would be nice to add one paragraph in the Discussion to give some general recommendations on the diet selection based on the data presented.

4.       For convenience, authors divided subjects into four groups based on the E-DII value. I suggest adding one graph to show subjects of CRC, colonic polyps and healthy control with original individual E-DII value. This would give a strong visual impression to the association of CRC risk and E-DII.

Reviewer 2 Report

In the manuscript „ Dietary Inflammatory Index, Obesity, and Incidence of Colorectal Cancer: Findings from Hospital-Based Case-Control Study in Malaysia” the Authors try to assess the relationship between the inflammatory impact of diet on the risk of colorectal cancer, as measured by the dietary inflammatory index score among obese and healthy weight individuals. This is a hospital-based case-control study conducted in two different cities in Peninsular Malaysia. The purposive sampling was conducted, which deserves recognition. This study fills the gap by providing new evidence on culture-specific dietary patterns in Malaysia that supports the intake of diets with greater pro-inflammatory potential with an increased incidence of colorectal cancer.

Generally, the manuscript provides valuable information. In the introduction, there is a brief description of the problem and the purpose of the study. The Authors very clearly described materials and methods. Moreover, the Authors have got approval from the Universiti Kebangsaan Malaysia Medical and Research Ethics Committee and the Human Research Ethics Committee of Universiti Sains Malaysia. The Authors used proper procedure in data analysis. The results are described well and concisely. The discussion is conducted in an interesting way, and the conclusions relate to the study.

However, I have some questions.

How exactly was the sample drawn? Could you explain?

Why did the FFQ questionnaire combine the “never” and “rarely” categories in one? There is a difference if someone never eats something and someone eats it rarely. Don't you think this could have affected the flawed results about consumption?

There are a lot of extensive and detailed tables in the article. I suggest moving some of them to supplementary materials.

Reviewer 3 Report

Thanks to submit the manuscript "Dietary Inflammatory Index, Obesity, and Incidence of Colorectal Cancer: Findings from Hospital-Based Case-Control

Study in Malaysia" to Nutrients.

research has evaluated the relationship between weight, disease conditions and disease and this relationship in a population is relevant research. However, the manuscript needs to be more readable and more didactic, such as making the results of the table more attractive.

keywords: please avoid words that make up the title.

Line#81: I think it's better to call the lean individual a non-obese individual

Line#115: Please add a brief explanation of why this cut line was used.

Line#150: It is unclear whether the modified questionnaire was also validated.

Line#180: as the index was not built or validated in this research, but it was only used as a tool, describe in general terms how to apply it and not how it was developed.

Table 2 needs to be separated to be readable.
